# Strain-Mediated Magneto-Electric Effects in Coaxial Nanofibers of Y/W-Type Hexagonal Ferrites and Ferroelectrics

Ying Liu [1,2], Peng Zhou [2], Bingfeng Ge [1,3], Jiahui Liu [1,3], Jitao Zhang [3], Wei Zhang [1], Tianjing Zhang [2] and Gopalan Srinivasan [1,*]

1   Physics Department, Oakland University, Rochester, MI 48309, USA; liuying.hube@outlook.com (Y.L.);
    BingfengGe@outlook.com (B.G.); Jiahui-Liu@outlook.com (J.L.); weizhang@oakland.edu (W.Z.)
2   Department of Materials Science and Engineering, Hubei University, Wuhan 430062, China;
    p_zhou@outlook.com (P.Z.); zhangtj@hubu.edu.cn (T.Z.)
3   College of Electrical and Information Engineering, Zhengzhou University of Light Industry,
    Zhengzhou 450002, China; zhang_jitao@outlook.com
*   Correspondence: srinivas@oakland.edu

**Abstract:** Nanofibers of Y- or W-type hexagonal ferrites and core–shell fibers of hexagonal ferrites and ferroelectric lead zirconate titanate (PZT) or barium titanate (BTO) were synthesized by electrospinning. The fibers were found to be free of impurity phases, and the core–shell structure was confirmed by electron and scanning probe microscopy. The values of magnetization of pure hexagonal ferrite fibers compared well with bulk ferrite values. The coaxial fibers showed good ferroelectric polarization, with a maximum value of 0.85 $\mu C/cm^2$ and 2.44 $\mu C/cm^2$ for fibers with BTO core–$Co_2W$ shell and PZT core–$Ni_2Y$ shell structures, respectively. The magnetization, however, was much smaller than that for bulk hexaferrites. Magneto-electric (ME) coupling strength was characterized by measuring the ME voltage coefficient (MEVC) for magnetic field-assembled films of coaxial fibers. Among the fibers with Y-type, films with $Zn_2Y$ showed a higher MEVC than films with $Ni_2Y$, and fibers with $Co_2W$ had a higher MEVC than that of those with $Zn_2W$. The highest MEVC of 20.3 mV/cm Oe was measured for $Co_2W$–PZT fibers. A very large ME response was measured in all of the films, even in the absence of an external magnetic bias field. The fibers studied here have the potential for use in magnetic sensors and high-frequency device applications.

**Keywords:** magnetoelectric; ferromagnetic; ferroelectric; nanofibers





## 1. Introduction

The magneto-electric (ME) effects in ferromagnetic–ferroelectric composites have been a topic of fundamental and technological interests in recent years [1–5]. The cross-coupling between the two ferroic phases is aided by mechanical deformation due to the piezoelectric effect in ferroelectricity and magnetostriction in the ferromagnetic phase [3,5]. Two types of ME effects are studied in such composites. These are the direct ME effect (DME), i.e., the response of the composite to an applied magnetic field that is measured as variations in the ferroelectric order parameters, and the converse ME effect (CME), which is the response of the composite to an applied electric field and measured in terms of changes in magnetic parameters [5]. Composites with a variety of ferromagnetic and ferroelectric phases have been investigated in the past, with a focus on composites in the form of bulk samples, bilayers, symmetric or asymmetric trilayers, nanobilayers, nanopillars, and nanofibers [3]. Measurement techniques for the strength of DME effects include the ME voltage coefficient and variations in the ferroelectric polarization and dielectric constant under an applied magnetic field H. For CME effects, one may measure the changes in the magnetization or the magnetic anisotropy field under an applied electric field E. Use of the composites as magnetic sensors, multiple-states memory devices, low-power spintronics, energy harvesting, and microwave devices has recently been investigated [6–9].

The most important ingredient for strong ME effects is efficient transfer of strain produced in a magnetic or electric field in one phase of the composite to another. Nanocomposites with a very high surface area-to-volume ratio, such as nanobilayers, nanopillars, and nanowires, are of interests in this regard [10–12]. Core–shell fibers are of particular importance for strong ME effects, since they are free of clamping due to substrates encountered in bilayers or the nanopillars on a substrate [13,14]. We have previously reported on ME effects in coaxial fibers of ferrites and ferroelectric lead zirconate titanate (PZT) and barium titanate (BTO) [15,16]. The fibers were made by electrospinning, and a variety of microscopy techniques showed the expected core–shell structure. Strong ME coupling was inferred from measurements of the ME voltage coefficient, induced polarization in H, and ferromagnetic resonance (FMR) measurements at 5–10 GHz with a scanning microwave microscope (SMM) under an applied E field [15–18].

This report focuses on ME effects in nanocomposites of W- or Y-type hexagonal ferrites and PZT or BTO, which are predicted to have strong ME coupling [14]. Hexagonal ferrites consist of spinel and hexagonal blocks and are classified into M-, U-, W-, X-, Y-, and Z-type based on the crystal structure, and they all have a large planar or uniaxial magnetocrystalline anisotropy field, as high as 33 kOe [19–22]. Since the anisotropy field acts as a built-in bias field, composites with hexaferrites are expected to show a large DME response under a zero external magnetic field and are ideally suitable for use as AC or DC magnetic field sensors. Studies have been conducted on ME coupling in composites with polycrystalline and single-crystal hexaferrites [23–25]. Layered composites of pure and Al-substituted single-crystal M-type hexaferrites and PZT have been reported to show strong direct and converse ME effects [23,25]. Recent studies of relevance to this report include those on core–shell nanofibers of spinel or M-type strontium hexaferrite and PZT/BTO [16,26–28]. The fibers prepared by electrospinning were found to show strong DME by measurements of the ME voltage coefficient and induced ferroelectric polarization under an applied H.

This work constitutes the first report of the synthesis of pure Y-type ($Zn_2Y$ and $Ni_2Y$) and W-type ($Zn_2W$ and $Co_2W$) nanofibers and coaxial composite fibers with PZT and BTO by electrospinning and an examination of the resultant ME coupling. Fibers of pure hexaferrites were free of impurities and characterization by magnetization, FMR, and magnetostriction yielded parameters in agreement with values for bulk ferrites. Fibers of PZT and BTO were synthesized by electrospinning, and ferroelectric order parameters were in agreement with values reported in previous studies [29–31]. Core–shell fibers of hexaferrite-BTO and hexaferrite-PZT prepared by electrospinning were annealed at 900–1200 °C, and scanning microwave microscopy images showed a core–shell structure free of any defects. Magnetic and ferroelectric order parameters for the composite fibers were smaller than for pure ferrite and ferroelectric fibers. The presence of a magnetocrystalline anisotropy was evident from FMR measurements. Strong ME coupling in the composite fibers was inferred from measurements of the low-frequency ME voltage coefficient (MEVC), and the fibers with $Co_2W$ showed the highest MEVC. A strong ME response under zero external magnetic bias was inferred from MEVC data for all of the fiber composites, and it was attributed to the magnetic anisotropy fields in the ferrite. Composite fibers could be of importance for applications in magnetic sensors and energy harvesting.

## 2. Materials and Methods

### 2.1. Fabrication of Pure Hexagonal Ferrite Nanofibers

Fibers of Y- and W-type ferrites were made by electrospinning and involved the following steps: first, sol for synthesis was made using nitrates of the constituents; second, the sol was loaded onto a syringe and dispensed through a stainless steel needle; finally, an electric field of 1.5 to 2 kV/cm was applied between the needle and a rotating aluminum drum that was used to collect the fibers. The sols for the synthesis of $Ba_2Ni_2Fe_{12}O_{22}$ ($Ni_2Y$), $Ba_2Zn_2Fe_{12}O_{22}$ ($Zn_2Y$), $BaCo_2Fe_{16}O_{27}$ ($Co_2W$), and $BaZn_2Fe_{16}O_{27}$ ($Zn_2W$) were made with the following nitrates: $Ni(NO_3)_2 \cdot 6H_2O$, $Zn(NO_3)_2 \cdot 6H_2O$, $Co(NO_3)_2 \cdot 6H_2O$, $Ba(NO_3)_2$, and

Fe(NO$_3$)$_3$·9H$_2$O. The nitrates totaling about 1–1.5 g, with the constituents proportional to their content in the ferrite, were dissolved in 7–9 mL of 2,5-dimethylfuran (DMF) at room temperature and stirred for 1–2 h. Then, poly(vinyl pyrrolidone) (PVP, MW~1,300,000) was added to the precursor solutions, which were stirred for another 12 h to obtain a homogeneous sol with PVP concentration of 10 wt. % for electrospinning.

The electrospinning chamber consisted of a syringe pump, a syringe with the sol, a needle (rame-hart Inc., needle JG 18, Succasunna, NJ, USA), a 20 kV power supply, and an Al-drum collector. DC voltage of 10–20 kV was applied between the needle and the drum at a distance of 10 cm from the tip of the needle so that electric field of 1.0–2.0 kV/cm was applied during electrospinning. The pump dispensed the sol at a rate of 0.1 mL/h, and the humidity inside the electrospinning chamber was kept at about 35–40%. The as-spun fibers were collected, dried in an oven at 50 °C for 24 h, and then annealed in air at 900–1200 °C. It was necessary to use heating and cooling rates of 0.5 °C/min to avoid disintegration of the fibers.

### 2.2. Preparation of Coaxial Nanofibers by Electrospinning

For coaxial nanofibers with hexagonal ferrite and ferroelectrics, sols of PbZr$_{0.52}$Ti$_{0.48}$O$_3$ (PZT) and barium titanate BaTiO$_3$ (BTO) were prepared using the procedure outlined in our previous work [16]. The sols of the ferrite and ferroelectric were loaded in two separate syringes that were mounted in a dual syringe pump. The sols were dispensed through a coaxial, dual-chamber stainless steel needle (rame-hart Inc., needle JG 17–20), and fibers generated under an electric field of 1–2 kV/cm were collected on a metal foil drum as well as on silicon substrates mounted on the drum. The fibers were dried in air at 50 °C for 24 h. Fibers of hexaferrite-BTO and hexaferrite-PZT were annealed at 1200 and 1000 °C, respectively, whereas fibers on silicon substrates were annealed at 900 °C.

### 2.3. Characterization of Nanofibers

Structural characterization of the fibers was carried out using a Bruker D8 advanced powder X-ray diffractometer (XRD) with Cu K$\alpha$ radiation. A scanning electron microscope (JEOL JSM 6510, JEOL Ltd., Akishima, Tokyo) and a scanning microwave microscope (Agilent Technology, Santa Clara, CA, United States) composed of a standard 5420 atomic force microscope (AFM) and a 25 GHz Vector Network Analyzer (Agilent Technologies, PNA) microwave source were used to investigate the structure and morphology the fibers. The magnetization of fibers was measured using a Quantum Design MPMS system and a Faraday susceptibility balance. Ferromagnetic resonance (FMR) measurements were carried out with a coplanar waveguide and an Agilent 67 GHz VNA. The magnetostriction was measured using a strain gage and a strain indicator. The magneto-electric (ME) effects in the core–shell fibers were measured by a ME voltage coefficient at 30 Hz in films of fibers assembled in a magnetic field. The films were assembled between two electrodes on a glass substrate by allowing a solution of fibers in alcohol orient and dry under a magnetic field of 3 kOe. The films were then subjected to an ac magnetic field of 1 Oe at 30 Hz and a static magnetic field H, and the ME voltage generated across the electrodes was measured as a function of H with a lock-in amplifier.

## 3. Results

### 3.1. Characterization of Y- and W-Type Hexagonal Ferrite Nanofibers

The XRD pattern obtained for the ferrite fibers annealed at 1200 °C are shown in Figure 1, and the expected powder diffraction peaks for the hexagonal ferrites from the available PDF data are also shown. It is noteworthy that peaks corresponding to any impurity phases are absent in the data for Ni$_2$Y and Zn$_2$W fibers. The peaks marked in red for Zn$_2$Y and Co$_2$W fibers were identified as BaFe$_{12}$O$_{19}$ (BaM), which formed during the high-temperature annealing. Similar impurities were reported in a previous work on Co$_2$W fiber when annealing was carried out at 1000 °C, but the impurities were absent, and a single-phase Co$_2$W was inferred when annealed at 1200 °C [32]. The comparison of

the XRD patterns for the fibers and the PDF standard shown in Figure 1 reveals that not all of the expected peaks can be observed in the fiber data, but some of the additional peaks were identified as impurity phases. In addition, factors such as stress on the nanofiber under test can lead to weakened peak intensities or deviations between the actual peak positions and reference values. This is evident, for example, in the XRD data of $Co_2W$ and $Zn_2W$ nanofibers in which a peak around 23 deg was observed for $Co_2W$ but not for $Zn_2W$.

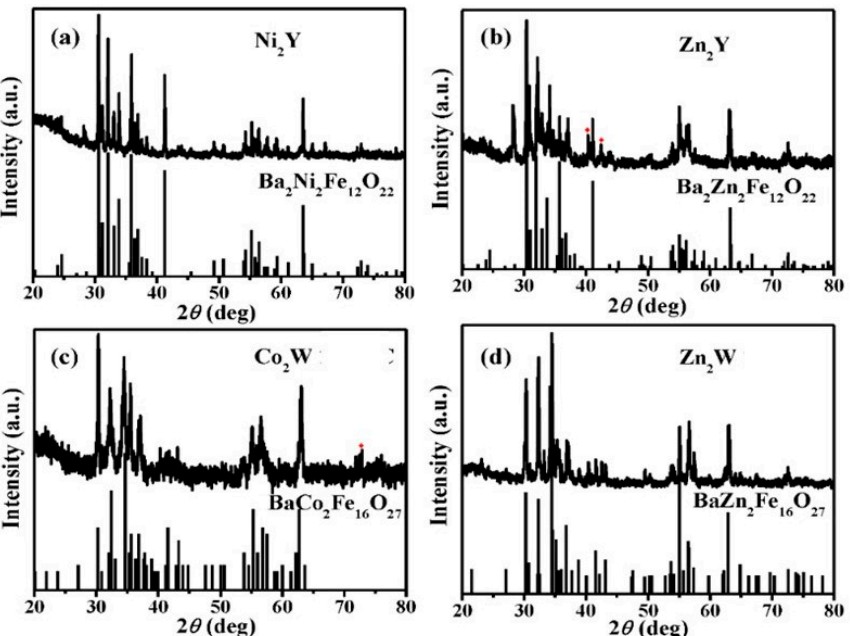

**Figure 1.** The XRD $\theta$–$2\theta$ pattern for fibers of (**a**) $Ni_2Y$, (**b**) $Zn_2Y$, (**c**) $Co_2W$, and (**d**) $Zn_2W$ annealed at 1200 °C. The peaks indicated by red dots in (**b**,**c**) belong to the impurity phase.

The morphology and diameter distribution of the as-spun nanofibers were determined by electron microscopy. A representative SEM image and diameter distribution data of $Zn_2Y$ fibers spun under E = 1.2 kV/cm are shown in Figure 2. The diameter distribution was calculated from measurements on 100 fibers. There are 65% of nanofibers with a diameter in the range of 120–180 nm. The fiber diameter was found to be a function of several factors, including sol viscosity, strength of the electric field applied during electrospinning, and humidity in the chamber. Figure 2 also shows the SEM image of fibers annealed at 1000 °C. A reduction in the fiber diameter that occurred with 65% of the fibers in the range of 80–120 nm was inferred from SEM images.

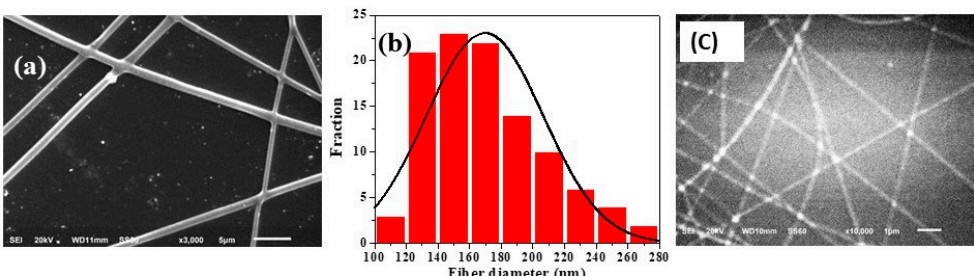

**Figure 2.** (**a**) SEM image of as-spun $Zn_2Y$ fibers, (**b**) fiber diameter distribution for as-spun fibers, and (**c**) SEM image of $Zn_2Y$ fibers annealed at 1000 °C.

The room temperature magnetization vs. magnetic field H data for the hexaferrite fibers annealed at 1200 °C are shown in Figure 3. All of the fibers showed magnetization saturation for H ranging from 5 kOe for $Ni_2Y$ to 20 kOe for the other three hexaferrites,

and the saturation magnetizations $M_s$ = 22.5, 66.8, 53.4, and 70.2 emu/g for $Ni_2Y$, $Zn_2Y$, $Zn_2W$, and $Co_2W$ fibers, respectively. The $M_s$ value for $Ni_2Y$ fiber compares well with the value for the polycrystalline counterpart, but for the $Co_2W$ fiber, $M_s$ is 10% higher than 60 emu/g reported for polycrystalline samples, which could be due to the BaM impurity phase that can be observed in the XRD data in Figure 1 [25].

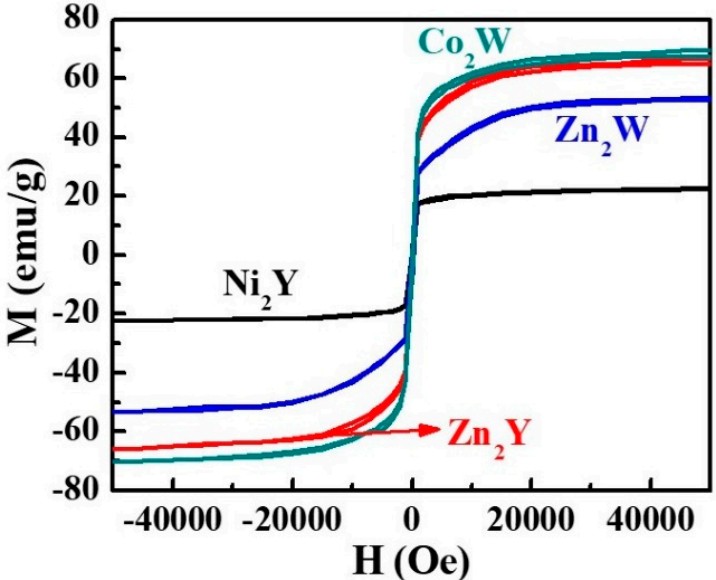

**Figure 3.** Room temperature magnetization versus H data for annealed fibers of $Ni_2Y$, $Zn_2Y$, $Zn_2W$, and $Co_2W$.

We carried out FMR measurements on discs of annealed fibers in order to obtain information on the nature and magnitude of the anisotropy field. The discs were placed in a coplanar waveguide microwave excitation structure and subjected to a static field H perpendicular to the sample plane. A vector network analyzer was used to record the scattering matrix $S_{11}$ as a function of frequency f, and FMR was seen as a dip in the $S_{11}$ vs. f data. Figure 4 shows the FMR in Y-type ferrites for a series of H values. The frequency width for FMR corresponds to a 3 dB linewidth $\Delta H$ of 1 to 2.5 kOe for $Ni_2Y$ and 1 to 4 kOe for $Zn_2Y$ for the frequency range 14–28 GHz. We were unable to record FMR for W-type ferrites most likely due to their very broad absorption profiles.

The resonance condition for H perpendicular to the sample plane is given by

$$f_r = \gamma \times (H - 4\pi \times M_{eff}) \tag{1}$$

where $\gamma$ is the gyromagnetic ratio and the effective saturation induction $4\pi M_{eff} = 4\pi M_s - H_a$, where $H_a$ is the anisotropy field. Figure 4 shows the data of resonance frequency $f_r$ as a function of H. The estimated $\gamma$ and $4\pi M_{eff}$ from linear fit to the data are $\gamma$ = 3.6 GHz/kOe and $4\pi M_{eff}$ = 2.22 kG for $Ni_2Y$ and 2.8 GHz/kOe and 2.19 kG for $Zn_2Y$. The magnetization in Figure 3 corresponds to $4\pi M_s$ = 1.5 kG and 2.85 kG for $Ni_2Y$ and $Zn_2Y$, respectively. Thus, the values of the anisotropy field $H_a$ are ~ 700 Oe for both Y-type ferrites, and $H_a$ is parallel to the sample plane for $Ni_2Y$ and is perpendicular to the sample plane for $Zn_2Y$. The FMR data in Figure 4, therefore, confirm the presence of an effective anisotropy field in the hexaferrite discs made of nanofibers.

We measured the magnetostriction $\lambda$ for the hexaferrites since the piezomagnetic coefficient q = d$\lambda$/dH is a key parameter that determines the strength of ME coupling in nanocomposites. Figure 5 shows data on the room-temperature magnetostriction $\lambda$ as a function of the static field H for hexaferrite fibers. Measurements were carried out on a disc of annealed fibers and with H applied parallel ($\lambda_{11}$) and perpendicular ($\lambda_{12}$) to the strain gauge. The hexaferrites all have a negative value for $\lambda_{11}$ and positive value of $\lambda_{12}$,

the $Ni_2Y$ fiber attained higher $\lambda$ than the $Zn_2Y$ fiber, and they both showed a tendency toward saturation for H ~ 3 kOe. The $Zn_2W$ and $Co_2W$ fibers showed the similar behavior, but the magnetostriction $\lambda_{11}$ did not show saturation at H ~ 3 kOe.

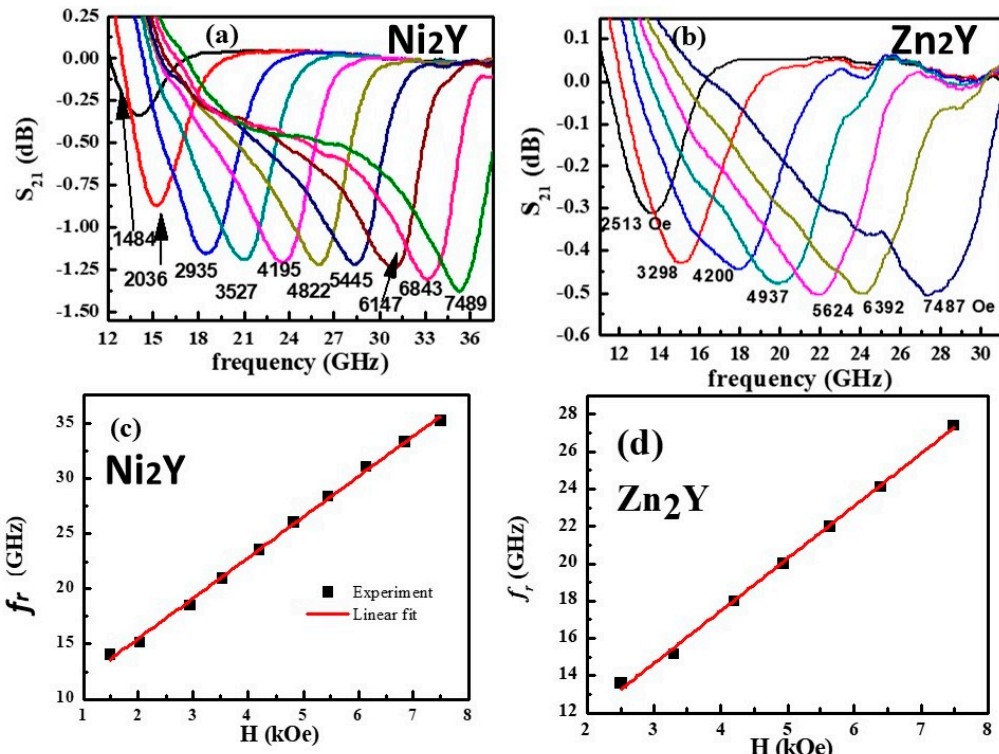

**Figure 4.** Profiles of scattering parameter $S_{11}$ versus frequency f showing ferromagnetic resonance (FMR) for a series of applied static fields H in Oe for discs of fibers of (**a**) $Ni_2Y$ and (**b**) $Zn_2Y$. The field H was applied perpendicular to the plane of the sample. (**c**,**d**) Resonance frequency $f_r$ as a function of H. The straight lines are linear fit to the $f_r$ vs. H data.

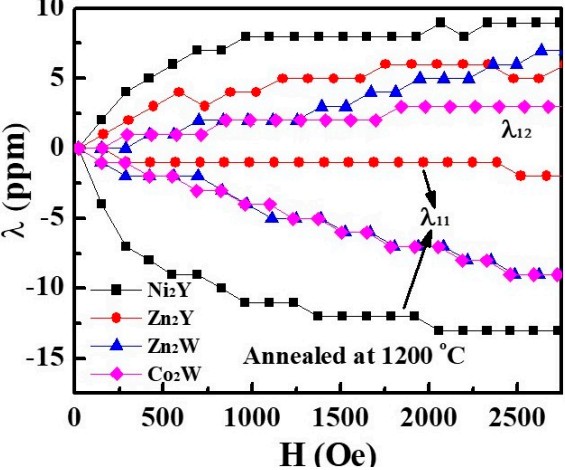

**Figure 5.** Parallel magnetostriction $\lambda_{11}$ and perpendicular magnetostriction $\lambda_{12}$ vs. H data measured in a disc of fibers of Y- and W-type hexagonal ferrites.

We prepared nanofibers of PZT and BTO and characterized them in terms of structural and ferroelectric order parameters. Details on the synthesis procedure and characterization are provided in [15,16]. The results of measurements of polarization P versus E on discs of fibers annealed at 900 °C showed maximum P of 2.5 $\mu C/cm^2$ and 0.4 $\mu C/cm^2$ for PZT and

BTO, respectively, (as shown in Figure S1 in the Supplement), and these values compare well with values reported for nanofibers [29,30].

### 3.2. Structural Characterization of Core–Shell Composite Fibers

Structural characterization of all of the core–shell fibers was carried out using X-ray diffraction (XRD). The XRD $\theta$–2$\theta$ patterns for the fibers of Y- and W-type hexagonal ferrites and ferroelectric PZT and BTO are shown in Figure 6. In Figure 6a, the data are for coaxial nanofibers of hexaferrites and BTO that were annealed at 1200 °C. Similar data for fibers of hexaferrites and PZT annealed at a lower temperature of 1000 °C are shown in Figure 6b. The lower annealing temperature was necessary to avoid loss of Pb from PZT, but the data indicate a small amount of BaM in the fibers due to the low annealing temperature [32].

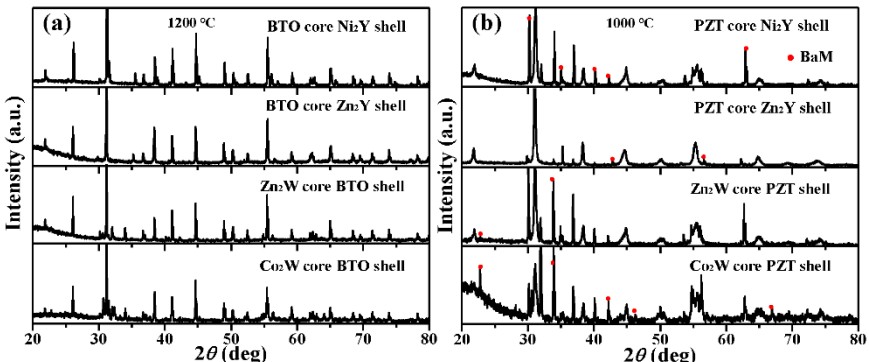

**Figure 6.** The XRD $\theta$–2$\theta$ patterns for (**a**) BTO core–Ni$_2$Y/Zn$_2$Y shell and Zn$_2$W/Co$_2$W core–BTO shell fibers annealed at 1200 °C, and (**b**) PZT core–Ni$_2$Y/Zn$_2$Y shell and Zn$_2$W/Co$_2$W core–PZT shell fibers annealed at 1000 °C. The red dot shows the peaks of impurity phase BaM.

In Figure 7, the core–shell structures of BTO core–Ni$_2$Y shell and Ni$_2$Y core–BTO shell fibers are clearly visible in the SEM images. The images are of fibers spun on silicon substrates and annealed at 900 °C. The contrast of the fibers distinctly shows the core and shell, with diameters of 108 and 124 nm for the BTO core–Ni$_2$Y shell and 120 nm and 147 nm for the PZT core–Zn$_2$Y shell, respectively. In addition, scanning microwave microscopy (SMM) was utilized for imaging the core–shell structure as shown in Figure 8.

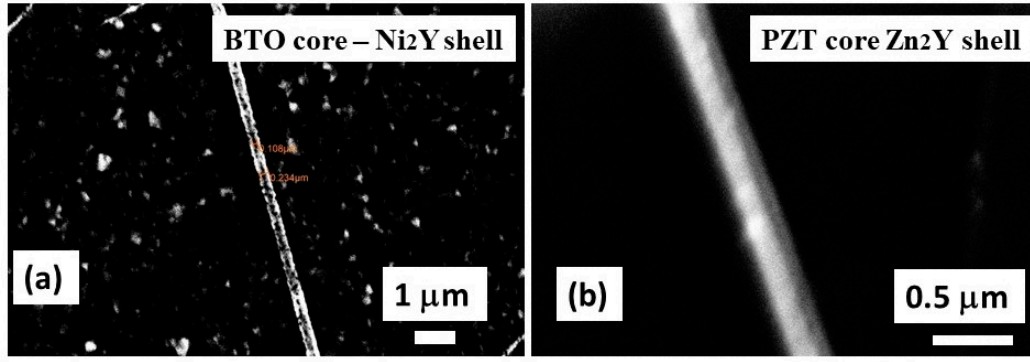

**Figure 7.** SEM images of (**a**) BTO core–Ni$_2$Y shell and (**b**) Ni$_2$Y core–BTO shell fibers spun under the electric field of 1.2 kV/cm and annealed at 900 °C.

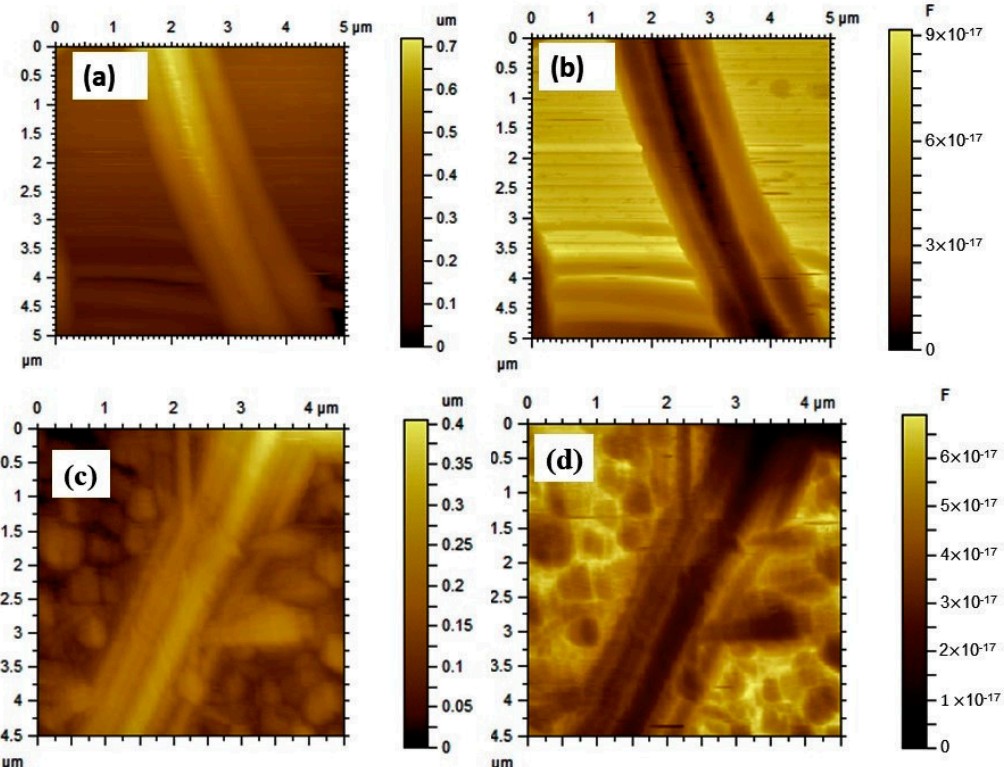

**Figure 8.** Scanning microwave microscopy (SMM) images recorded at 15.6 GHz showing (**a**) topography and (**b**) capacitance for a fiber with a BTO core and Zn$_2$Y shell. Similar topography and capacitance images of a fiber with a Zn$_2$Y core and PZT shell are shown in (**c,d**), respectively.

The SMM images of fiber topography and capacitance at 15.6 GHz for fibers with BTO core–Zn$_2$Y shell and Zn$_2$Y core–PZT shell structures in the figure clearly show the core and shell. Both are of uniform dimeter with a clear interface free of any distortion. (Additional SEM and SMM images are shown in Figures S2–S5 in the Supplementary Materials).

### 3.3. Ferroic Order Parameters for Core–Shell Composite Fibers

The room-temperature magnetization and polarization measurements for all of the coaxial nanofibers were performed using a Faraday balance and a Ferroelectric Tester. Representative magnetic hysteresis of hexaferrite-BTO and hexaferrite-PZT coaxial fibers are shown in Figure 9a,b. The magnetization of all of the coaxial fibers was not saturated for fields H = ±3 kOe. The coercive field and magnetization of the coaxial fibers of hexaferrite-PZT are higher than those of hexaferrite-BTO. The magnetization values of 8.5 emu/g and 6.08 emu/g for the PZT core–Co$_2$W shell and Co$_2$W core–BTO shell fibers, respectively, are higher than for the other fiber composites. (Figures S6–S8 show additional M vs. H data for the fibers). Ferroelectric polarization was measured on nanofibers that were pressed into a thin disk and coated with platinum electrodes. Representative ferroelectric hysteresis loop data of P vs. E are shown in Figure 9c,d. The maximum P-value for hexaferrite-PZT fibers is about 2 $\mu$C/cm$^2$ at 1 Hz, and ~1 $\mu$C/cm$^2$ for hexaferrite-BTO coaxial fibers at 100 Hz. For E = 11 kV/cm$^2$, remnant polarization has the highest value of P$_r$ = 0.294 $\mu$C/cm$^2$ for the BTO core–Co$_2$W shell fiber and P$_r$ = 0.381 $\mu$C/cm$^2$ for Co$_2$W core–PZT shell fiber. (Additional P vs. E data are shown in Figures S9–S11 in the Supplementary Materials).

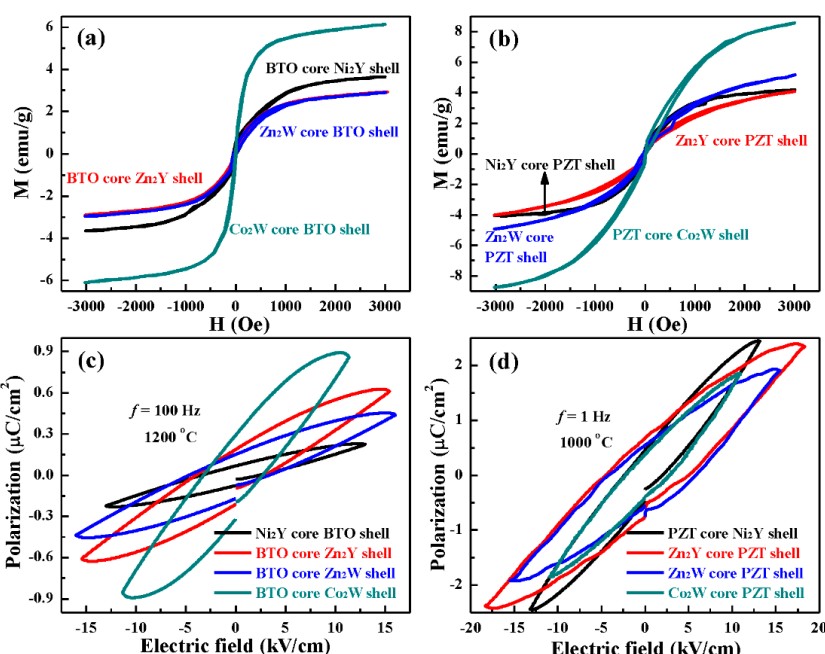

**Figure 9.** The room-temperature magnetic hysteresis loops of (**a**) BTO core–$Ni_2Y$/$Zn_2Y$ shell and $Zn_2W$/$Co_2W$ core–BTO shell fibers, and (**b**) $Ni_2Y$/$Zn_2Y$/$Zn_2W$ core–PZT shell and PZT core–$Co_2W$ shell fibers. Polarization vs. electric field for (**c**) $Ni_2Y$ core–BTO shell and $Zn_2Y$/$Zn_2W$/$Co_2W$ core–BTO shell fibers at the frequency of 100 Hz, and (**d**) PZT core–$Ni_2Y$ shell and $Zn_2Y$/$Zn_2W$/$Co_2W$ core–PZT shell fibers at the frequency of 1 Hz.

Ferromagnetic resonance (FMR) spectra were obtained for the fibers for additional magnetic characterization. Figure 10 shows the FMR profiles of the first derivative of the power absorbed (dP/dH) vs. H for $Zn_2Y$ core–BTO shell fibers pressed into a disk and annealed at 1200 °C. The fiber disk was placed in a coplanar waveguide with the applied magnetic field H in the plane of the disk. The FMR profiles for frequencies f = 5 to 9 GHz are shown in Figure 10a. The resonance field increased from 1500 to 2800 Oe with an increase in frequency from 5 to 9 GHz, and the FMR linewidth for $Zn_2Y$ core–BTO shell fiber was in the range of 280–360 Oe.

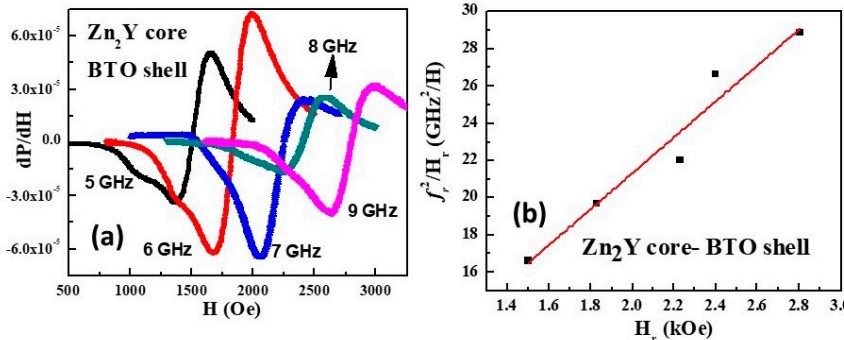

**Figure 10.** (**a**) The ferromagnetic resonance spectra for a disk of $Zn_2Y$ core–BTO shell fibers. The static field H was applied parallel to the sample plane. (**b**) Data on resonance frequency and field from profiles in (**a**) and linear fit of the data to Equation (2).

The condition for resonance for in-pane magnetic field is given by

$$f_r = \gamma \times H_r^{1/2} \times (H_r + 4\pi \times M_{eff})^{1/2} \qquad (2)$$

In Equation (2), $f_r$ is the resonance frequency and $H_r$ is the resonance field. Figure 10b shows data on $f_r^2/H_r$ as a function of $H_r$ obtained from FMR profiles as shown in

Figure 10a. A linear fit of the data to Equation (2) yielded values of $\gamma$ = 3.04 GHz/kOe and $4\pi M_{eff}$ = 200 G. The $\gamma$-value is 7% higher than the value obtained from FMR for pure $Zn_2Y$ fibers (Figure 4). An anisotropy field of 100 Oe was estimated from the effective saturation induction $4\pi M_{eff}$ and $4\pi M_s$ = 100 G (Figure S5 in the supplement). Similar FMR data for fibers of $Zn_2W$ and BTO are shown in the Figure S11, and the estimated anisotropy field was 150 Oe from $4\pi M_{eff}$ and magnetization data presented in Figure S6.

### 3.4. ME Characterization of Ferrite Core–Ferroelectric Shell Composite Fibers

Results regarding the strengths of the magneto-electric effect in the coaxial hexaferrite and ferroelectric fibers are presented here via the measurement of the magneto-electric voltage coefficient (MEVC). Annealed coaxial fibers were assembled into a 3D film with a thickness of 1 to 2 mm between two copper plates under an applied magnetic field. The films were subjected to a DC field H and an AC magnetic field $h_{ac}$ = 1 Oe at 30 Hz. The ME voltage V between the Cu electrodes was measured to estimate MEVC = $V/(t\ h_{ac})$ as a function of H. Data on the MEVC were obtained for both fields H and $h_{ac}$ parallel to each other and either parallel or perpendicular to the plane of the film.

Figure 11 shows MEVC vs. H data for composites with cores of Y-type ferrites and shells of either BTO or PZT. The key features are as follows: (i) For films with $Ni_2Y$, a large ME response with MEVC = 9 to 10 mV/cm Oe at zero-bias was observed for films with both BTO and PZT. (ii) With the application of an external static field H, the MEVC increased with H for films with $Ni_2Y$–BTO. (iii) There was no significant difference in the overall ME response for films with $Ni_2Y$ and PZT or BTO. (iv) Films with $Zn_2Y$ core also showed a large ME response at the zero bias field and had a higher MEVC than that of the fibers with $Ni_2Y$ cores. (v) Fibers with $Zn_2Y$ cores and a BTO shell had the highest MEVC amongst the samples with Y-type ferrite. (vi) Finally, for samples with $Zn_2Y$, the MEVC decreased with increasing H, and the ME response was stronger for out-of-plane H than for in-plane H. For films with $Ni_2Y$, however, the in-plane MEVC was higher than for the out-of-plane H.

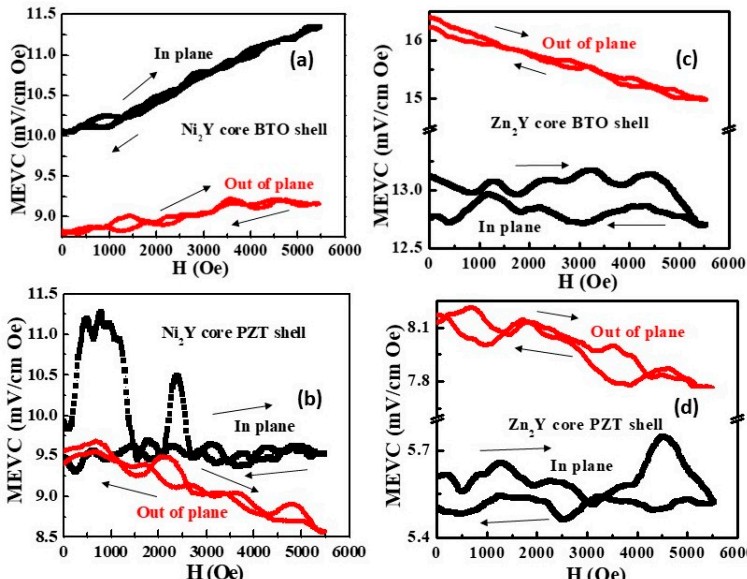

**Figure 11.** The room-temperature magneto-electric voltage coefficient (MEVC) as a function of applied field H for 3D films of ferrite cores and ferroelectric shells. Data are for H and $h_{ac}$ parallel to each other and either parallel or perpendicular to films of (**a**) $Ni_2Y$–BTO, (**b**) $Ni_2Y$–PZT, (**c**) $Zn_2Y$–BTO, and (**d**) $Zn_2Y$–PZT. The fibers with BTO were annealed at 1200 °C and assembled into a 3D film in a magnetic field. Fibers with PZT were annealed at 1000 °C. Arrows indicate data for increasing and decreasing H.

Similar MEVC data for composites with W-type cores and ferroelectric shells are shown in Figure 12. MEVCs at zero bias range from a minimum of 8 mV/cm Oe for fibers with $Zn_2W$ to a maximum of 12.5 mV/cm Oe for films with $Co_2W$ cores. Films with $Zn_2W$ showed an increase in the MEVC with increasing H, but samples with $Co_2W$ cores showed weakening of the ME response with increasing H. For fibers with $Zn_2W$, the ME interactions were stronger for out-of-plane H than for in-plane H. For films with $Co_2W$, however, the highest MEVC was measured for in-plane H, and the overall ME response was of the same strength for both kinds of ferroelectric shells.

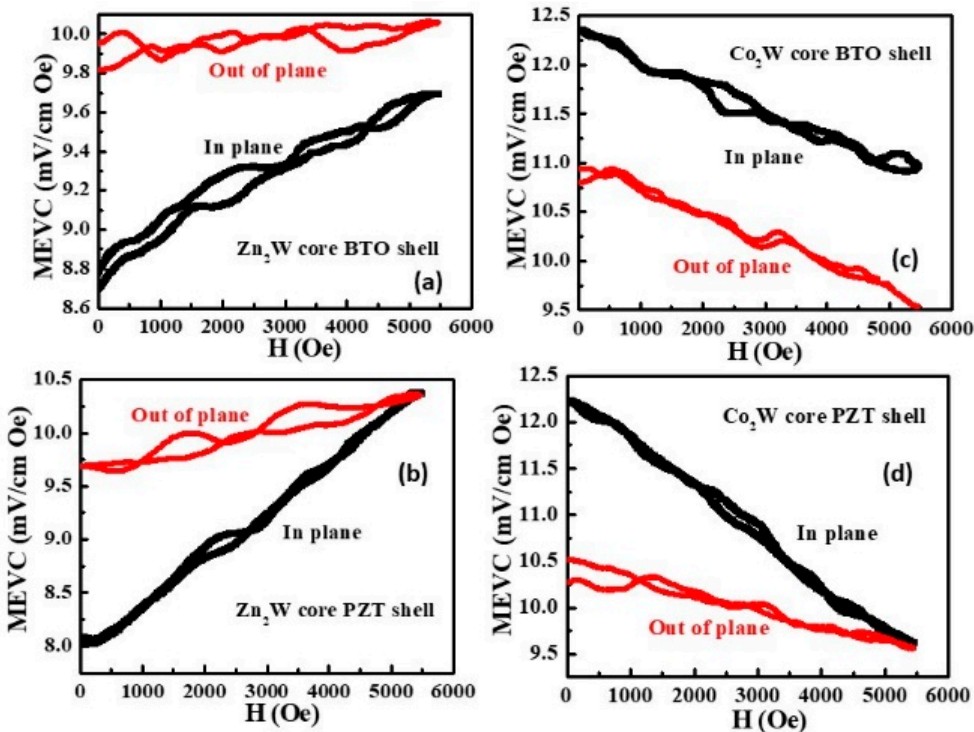

**Figure 12.** Data as shown in Figure 11 for 3D films with fibers of (**a**) $Zn_2W$ and BTO, (**b**) $Zn_2W$ and PZT, (**c**) $Co_2W$ and BTO, and (**d**) $Co_2W$ and PZT.

### 3.5. ME Characterization of Ferroelectric Core–Ferrite Shell Composite Fibers

The ME effects in films of fibers with BTO or PZT core–ferrite shell structures are examined in this section. Figure 13 shows data on the MEVC vs. H for fibers with shells of Y-type ferrite. The highest MEVC was measured for films with BTO core–$Zn_2Y$ shell structures and the lowest for those with PZT core–$Zn_2Y$ shell structures. The ME response weakened with increasing H for out-of-plane fields. Such behavior was not observed for in-plane fields. Hysteresis was evident for MEVC vs. H data for all of the films except for PZT core–$Zn_2Y$ shell films.

The static field H dependence of the ME voltage coefficient for films of ferroelectric shells and cores of W-type ferrites is shown in Figure 14. The most significant inference from the data is the highest MEVC of 20.2 mV/cm Oe measured for in-plane fields for films of PZT and $Co_2W$. The strengths of the ME coupling are comparable to those of films with BTO cores, but films with fibers of PZT cores have a higher MEVC than those of fibers with BTO cores. The hysteresis in the MEVC data in Figure 14 is notably small for the films with W-type ferrite shells.

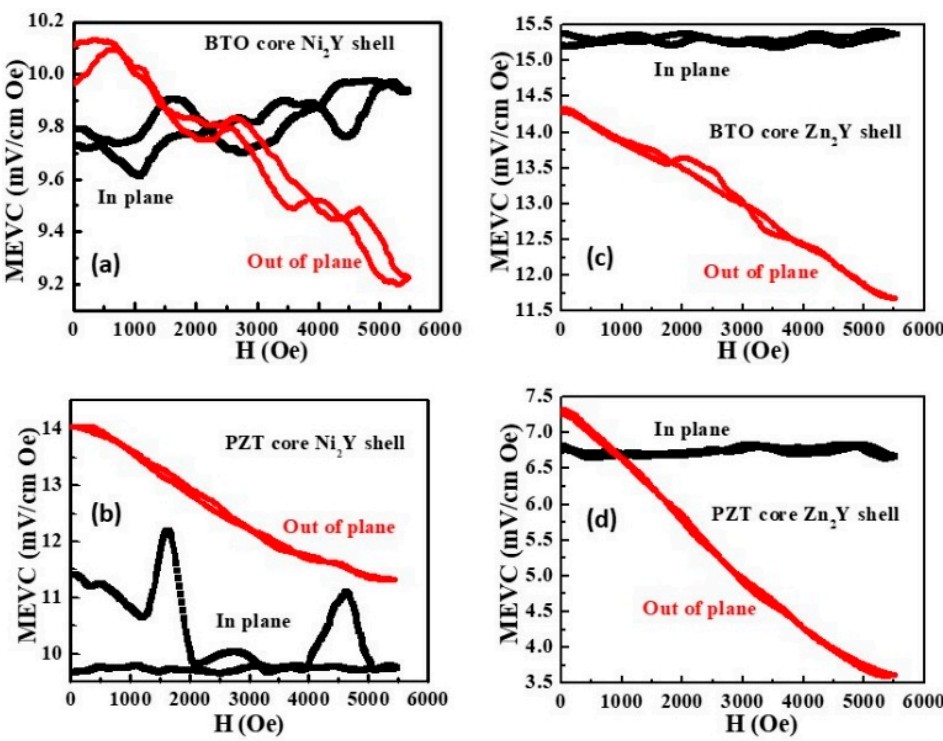

**Figure 13.** MEVC vs. H data for films of ferroelectric core and shells of Y-type hexaferrites. (**a**) BTO and Ni$_2$Y, (**b**) PZT and Ni$_2$Y, (**c**) BTO and Zn$_2$Y, and (**d**) PZT and Zn$_2$Y.

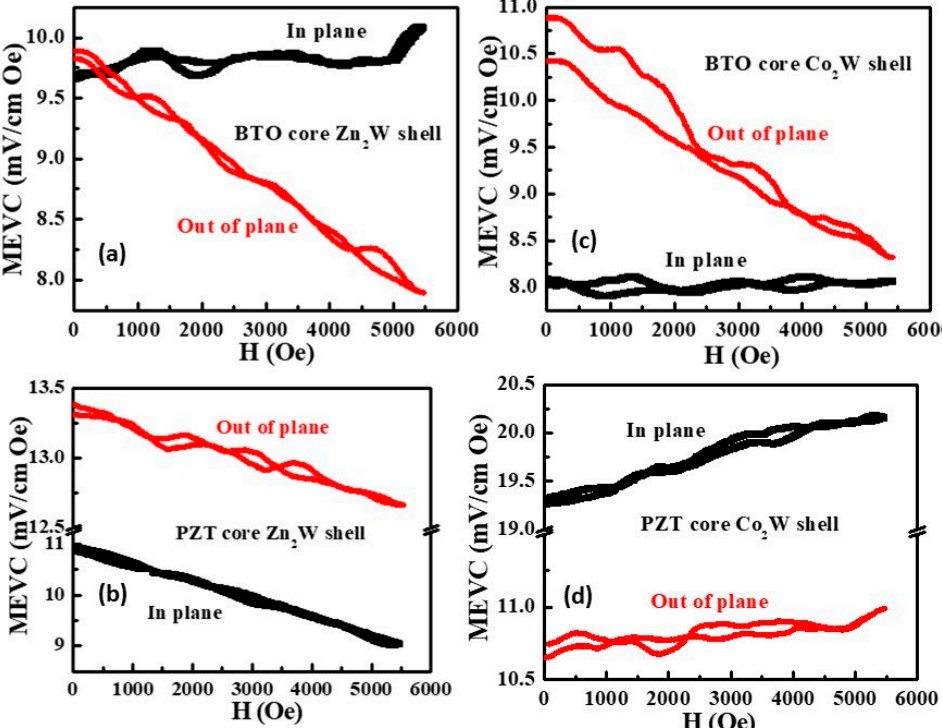

**Figure 14.** Data as shown in Figure 13 for films of fibers with ferroelectric core and W-type ferrite shells. (**a**) BTO and Zn$_2$W, (**b**) PZT and Zn$_2$W, (**c**) BTO and Co$_2$W, and (**d**) PZT and Co$_2$W.

## 4. Discussions

Y- and W-type hexaferrite fibers and the hexaferrite-BTO and hexaferrite-PZT coaxial fibers were successfully synthesized via electrospinning. The Y- and W-type hexaferrite fibers were annealed at 1200 °C. The XRD data in Figure 1 show no impurity phases in

Ni$_2$Y and Zn$_2$W fibers, but a small amount of BaM was observed to be present in Zn$_2$Y and Co$_2$W. The fiber diameters calculated from the SEM images of all of the as-spun fibers are in the range of 110–310 nm. Structural characterization of the hexaferrite-BTO coaxial fibers annealed at 1200 °C did not show any impurity phases. Composite fibers with PZT, however, were annealed at 1000 °C due to the volatility of Pb at high temperatures, and XRD data show some peaks of BaFe$_{12}$O$_{19}$ (BaM) because of the low annealing temperature. The SEM and SMM images of annealed coaxial fibers (Figures 7 and 8) clearly show the core–shell structure. The diameter of the ferrite core and the thickness of the shell were found to be much smaller than the ferroelectric-phase diameter and shell thickness. Therefore, the volume fraction for the ferrite in annealed fibers was notably small.

Room-temperature magnetization and magnetostriction of annealed Ni$_2$Y, Zn$_2$Y, Zn$_2$W, and Co$_2$W fibers were measured and compared with those of polycrystalline hexaferrites. There was a small increase in the magnetization of Zn$_2$Y and Co$_2$W compared to their polycrystalline counterparts due to the BaM impurity phase, but the magnetostriction of hexaferrite fibers was smaller than that of bulk polycrystalline ferrites [19,26,33]. Hexagonal ferrites have a large magnetocrystalline anisotropy field, and most Y-type hexaferrites have planar magnetic anisotropy at room temperature, [19,21,34] while M- and W-type hexaferrites have large uniaxial anisotropy along the c-axis, except for Co$_2$W, which has magnetic anisotropy in a cone at an angle from the c-axis [19]. Zn$_2$Y has lower planar anisotropy than that of Ni$_2$Y [34]. The ferromagnetic resonance measurements on fiber discs of Y-type ferrites yielded γ-values in agreement with bulk values and an anisotropy field of 700 Oe for both Ni$_2$Y and Zn$_2$Y.

Magnetization of the hexaferrite–ferroelectric coaxial fibers was measured to be 5 to 20 times smaller than that of pure hexaferrite fibers due to the low volume fraction for the ferrites in the composites. Discs of fibers with Zn$_2$Y core–BTO shell and Zn$_2$W core–BTO structures showed good FMR spectra at frequencies of 5–9 GHz with the applied magnetic field parallel to the plane of the fiber disk (Figure 10 and Figure S11). The FMR linewidth is in the range 280 to 360 Oe for the Zn$_2$Y core–BTO shell fiber, which is even smaller than the reported line widths for YIG–NKN core–shell fiber [35]. An anisotropy field of 100 to 150 Oe in the composite fibers was estimated from FMR measurements and is a factor 5 to 7 smaller than the values for pure hexaferrite fibers. The value of ferroelectric polarization for coaxial fibers with BTO is 10 to 40 times smaller than the value of 9.6 μC/cm$^2$ for pure BTO fibers, but it was observed to be much higher than the value of polarization for SrM core–BTO shell fibers [16]. Hexaferrite-PZT coaxial fibers have ferroelectric parameters comparable to those of pure PZT fibers (Figure S1). The reduction in the values of polarization for coaxial fibers compared to those of bulk BTO and PZT could be due to the leakage current expected with low-resistivity hexaferrite in composite fibers [16,18].

Magnetoelectric coupling strengths were measured for all of the hexaferrite-BTO and hexaferrite-PZT coaxial fibers by obtaining data on the MEVC under an AC field of 1 Oe at 30 Hz and for the applied DC magnetic fields up to 5.5 kOe. Measurements were conducted for H and h$_{ac}$ parallel to each other and either applied parallel or perpendicular to the film plane. The most significant finding from the data in Figures 11–14 is the large MEVC at H = 0, which could be attributed to the anisotropy field in the ferrite that acts as a built-in bias field. The low-frequency ME coefficient in the composite is directly proportional to the product of the piezomagnetic coefficient q = dλ/dH (where λ is the magnetostriction) and the piezoelectric coefficient d. In most of the ferromagnetic metals and spinel ferrites, λ and q ≈ 0 at H = 0, resulting in MEVC = 0 at H = 0. The ME response under zero external bias was reported in several systems, including layered composites with spinel ferrites and hexagonal ferrites [36,37]. In these composite systems, the remnant magnetization or the magnetocrystalline anisotropy acts as a bias field, and, therefore, a nonzero q-value and MEVC can be obtained. In the present study, we estimated an anisotropy field H$_a$ of 700 Oe in pure hexagonal ferrite fibers from FMR measurements. In the core–shell fibers,

FMR data yielded a smaller $H_a$ of 150–200 Oe that acts essentially as the bias field and results in a nonzero MEVC for H = 0.

The zero-field MEVC in Figure 11 for films with a Y-type ferrite core and ferroelectric shell range from a minimum of 5.5 mV/cm Oe to a maximum of 16 mV/cm Oe, with films with a BTO shell showing a higher MEVC than that of films with a PZT shell. There is no significant change in the MEVC at H = 0 when the core and shell materials are interchanged, as shown in the data presented in Figure 13. For films with fibers of a W-type ferrite core and ferroelectrics, the spread in MEVC values for H = 0 is rather small, from 8 to 12.5 mV/cm Oe (Figure 12). However, a substantial increase in the zero-field MEVC is measured when the core and shell materials are interchanged, as shown in Figure 14. A 30–60% increase can be achieved, especially in the case of films with PZT used for the ferroelectric phase. The zero-field MEVC is the highest for films with fibers with a PZT core and $Co_2W$ shell.

With the application of an external bias field H, the variation in the MEVC is due to H dependence of the piezomagnetic coefficient q. The data presented in Figures 11–14 show variations of 5 to 10% for in-plane fields and a much higher variation of 5 to 50% for out-of-plane fields when H is increased from 0 to 5.5 kOe. The rapid decrease in the MEVC with H for the field perpendicular to the film plane is due to a higher demagnetization field compared to in-plane fields.

The ME response of the composites is a function of material parameters and the volume fraction for the two phases [25]. In addition, the MEVC will also be a function of the distribution in the diameters and lengths of the fiber's core and shell. In spite of the anticipated dependence of the MEVC on such fiber parameters, the data in Figures 11 and 13 do not show a notable difference in the MEVC when the core and shell are interchanged for films with Y-type ferrite and ferroelectrics. One may draw a similar inference for films with W-type ferrite and ferroelectrics, except for films with fibers with a PZT core and $Co_2W$ shell with the highest MEVC. The MEVC vs. H data show hysteresis for all of the systems studied, and particularly for fibers with $Ni_2Y$ and PZT.

One may compare results of this study with the ME response of layered composites of hexagonal ferrites and ferroelectrics. In our recent study on symmetric trilayer composites, we reported a maximum MEVC of 100 mV/cm Oe for $Ni_2Y$–PZT and 26 mV/cm Oe for $Co_2W$–PZT [25]. The fibers with a PZT core–$Ni_2Y$ shell structure had a maximum MEVC of 14 mV/cm Oe, and for PZT core–$Co_2W$ shell fibers, the highest MEVC = 20.3 mV/cm·Oe [25]. The strength of ME interactions is expected to weaken in fibers due to several factors, including magnetic and electric dipole–dipole interactions between fibers in the assembled films, porosity, and leakage currents through the ferrite core or shell.

Finally, we compared the MEVC of hexaferrite–ferroelectric fibers with results for SrM–ferroelectric and spinel ferrite–ferroelectric fibers. For SrM–PZT fibers, the MEVC was measured for 2D and 3D films, with the 3D films showing a higher MEVC (14–22 mV/cm·Oe) than that of 2D films (3.2–3.8 mV/cm·Oe) [16]. For fibers of nickel ferrite with BTO, the MEVC was very small, on the order of 0.5 mV/cm Oe [18]. Thus, the fibers of Y- and W-type hexagonal ferrites and PZT and BTO show the highest MEVC reported to date for coaxial ferrite–ferroelectric nanofibers. The MEVC values, however, are smaller than those of the laminate counterparts made of polycrystalline ferrite–ferroelectric fibers [25]. In spite of the low MEVC values, these nanofibers are of importance for applications in miniature magnetic sensors for medical imaging, RF sensors, and energy harvesting [38]. Efforts such as optimization of the ME response via enhancement of the resistivity of ferrite by annealing fibers in an oxygen-rich atmosphere to eliminate the divalent Fe in order to decrease the leakage currents are recommended.

## 5. Conclusions

Nanofibers of Y- and W- type hexagonal ferrites and composite core–shell fibers with ferroelectric PZT and BTO were synthesized by electrospinning and were characterized in terms of structural and ferroic parameters. Fibers of pure hexaferrites had a diameter of

100–300 nm before annealing, and the magnetic order parameters of annealed fibers were in agreement with the bulk polycrystalline values. Electron microscopy and SPM images of ferrite–ferroelectric composite fibers showed the expected core–shell structure. The magnetization of the composite was much smaller than the value for pure hexaferrite due to a small volume fraction of the ferrites in the fibers. The magneto-electric coupling strength of the coaxial fibers was determined by measurements of the low-frequency MEVC. All of the magnetic field-assembled 3D films of the composite fibers showed evidence of strong ME coupling at a zero external field due to the presence of a magnetocrystalline anisotropy field. The highest MEVC of 20.3 mV/cm·Oe was measured for films of fibers of PZT core–$Co_2W$ shell. The highest MEVC for composites with Y-type ferrite was measured for films of $Zn_2Y$ core–BTO shell fiber. The hexagonal ferrite–ferroelectric nanofibers studied here show one of the highest ME coupling strengths for nanocomposites and are of interest for application in miniature magnetic sensors, energy harvesting, and microwave devices.

**Supplementary Materials:** The following are available online at https://www.mdpi.com/article/10.3390/jcs5100268/s1, Figure S1: XRD and P vs E data for nanofibers of PZT and barium titanate (BTO); Figure S2: SEM images for core-shell nanofibers; Figure S3: SMM images for core-shell fibers; Figure S4: SMM capacitance images for core-shell fibers; Figure S5: M vs H data for core-shell fibers; Figure S6: M vs H data for core-shell fibers; Figure S7: M vs H data for core-shell fibers; Figure S8: P vs E data for core-shell fibers; Figure S9: P vs E data for core-shell fibers; Figure S10: P vs E data for core-shell fibers; Figure S11: FMR profiles for core-shell fibers.

**Author Contributions:** Data curation, G.S.; formal analysis, Y.L., P.Z., and G.S.; funding acquisition, G.S., J.Z., W.Z. and T.Z.; investigation, Y.L., P.Z., B.G., and J.L.; methodology, J.Z. and W.Z.; project administration, T.Z.; supervision, W.Z.; writing—original draft, G.S. All authors have read and agreed to the published version of the manuscript.

**Funding:** The research conducted at Oakland University was supported by grants from the National Science Foundation (DMR-1808892, ECCS-1923732) and the Air Force Office of Scientific Research (AFOSR) Award No. FA9550-20-1-0114. Ying Liu was supported by a fellowship from the Chinese Scholarship Council. The research conducted at Hubei University was supported by the China Postdoctoral Science foundation (No. 2020M672315) and the Program of Hubei Key Laboratory of Ferro- & Piezoelectric Materials and Devices (No. K202013). The research conducted at the Zhengzhou University of Light Industry was supported by the National Natural Science Foundation of China (NSFC) Grant No.61973279.

**Institutional Review Board Statement:** Not applicable.

**Informed Consent Statement:** Not applicable.

**Data Availability Statement:** Data are available from corresponding author upon reasonable request.

**Conflicts of Interest:** The authors declare no conflict of interest.

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
