# Peer review of "Strain-Mediated Magneto-Electric Effects in Coaxial Nanofibers of Y/W-Type Hexagonal Ferrites and Ferroelectrics"

_jcs, doi:10.3390/jcs5100268_

Round 1

Reviewer 1 Report

The authors present an interesting and detailed study on magnetic properties of pure Y/W hexaferrite fibers and multiferroic properties of their composite core-shell nanofibers with ferroelectric BTO/PZT. I think the results are remarkable and the methods employed are both sophisticated and reliable, which can be noted from the characterizations and measurements obtained, as well as the clarity of the key phenomena. The pure-phase hexagonal ferrite and core-shell structured hexagonal ferrite-ferroelectric nanofibers with good ferroelectric and ferromagnetic properties were successfully synthesized. The core-shell nanofibers of hexagonal ferrite-ferroelectric studied here show high ME coupling strengths, and providing a new path for the development of applications such as miniature magnetic sensors, energy harvesting, and microwave devices.

I think the article is suitable for publication in Journal of Composites Science, but there are some points which I think should be explained in a little more detail to increase the clarity of the paper before publication.

  1. Please describe briefly the new methods or results that are of sufficient interest to warrant this article's publication in Journal of Composites Science.
  2. The MEVC for 3D films of coaxial fibers have a highest value of 22 mV/cm·Oe than other reported coaxial ferrite-ferroelectric nanofibers, but the value is much smaller than that of the laminate composites. What are the advantages of coaxial nanofibers compared to laminate composites and how can they be optimized?
  3. Please check the minor issues in the manuscript, such as the subscript for Zn2Y on page 15, and providing high-quality images for Figures 4.

Author Response

Response to comments by Reviewer 1

We are grateful to the reviewer for the comments.  The manuscript has been revised by taking into consideration all of the comments. Our response to specific comments is given below.

“The authors present an interesting and detailed study on magnetic properties of pure Y/W hexaferrite fibers and multiferroic properties of their composite core-shell nanofibers with ferroelectric BTO/PZT. I think the results are remarkable and the methods employed are both sophisticated and reliable, which can be noted from the characterizations and measurements obtained, as well as the clarity of the key phenomena. The pure-phase hexagonal ferrite and core-shell structured hexagonal ferrite-ferroelectric nanofibers with good ferroelectric and ferromagnetic properties were successfully synthesized. The core-shell nanofibers of hexagonal ferrite-ferroelectric studied here show high ME coupling strengths, and providing a new path for the development of applications such as miniature magnetic sensors, energy harvesting, and microwave devices.

I think the article is suitable for publication in Journal of Composites Science, but there are some points which I think should be explained in a little more detail to increase the clarity of the paper before publication.”

  1. Please describe briefly the new methods or results that are of sufficient interest to warrant this article's publication in Journal of Composites Science.

Response:  To our knowledge this work is the first report on the synthesis of coaxial nanofibers of Y- and W-type hexagonal ferrites and characterization of the magneto-electric interactions.  We added the following paragraph to the Introduction part.

This work constitutes the first report on the synthesis by electrospinning of pure Y- type (Zn2Y and Ni2Y) and W-type (Zn2W and Co2W) nanofibers and coaxial composite fibers with PZT and BTO and studies on ME coupling. Fibers of pure hexaferrites were free of impurities and characterization by magnetization, FMR, and magnetostriction yielded parameters in agreement with values for bulk ferrites. Fibers of PZT and BTO were synthesized by electrospinning and the ferroelectric order parameters were in agreement with values reported in the past works [29-31]. Core-shell fibers of hexaferrite-BTO and hexaferrite-PZT prepared by electrospinning were annealed at 900-1200 ℃ and scanning microwave microscopy images showed the core and shell structure free of any defects. Magnetic and ferroelectric order parameters for the composite fibers were smaller than for pure ferrite and ferroelectric fibers. The presence of a magnetocrystalline anisotropy was evident from FMR measurements. Strong ME coupling in the composite fibers was inferred from measurements of the low frequency ME voltage coefficient (MEVC) and the fibers with Co2W showed the highest MEVC. A strong ME response under zero external magnetic bias was inferred from MEVC data for all the fiber composites that was attributed to the magnetic anisotropy fields in the ferrite.  The composite fibers could be of importance for applications in magnetic sensors and energy harvesting.

  1. The MEVC for 3D films of coaxial fibers have a highest value of 22 mV/cm·Oe than other reported coaxial ferrite-ferroelectric nanofibers, but the value is much smaller than that of the laminate composites. What are the advantages of coaxial nanofibers compared to laminate composites and how can they be optimized?

Response:  It is true that the nanocomposites studied in this work have ME coefficients smaller than the laminate composites made of polycrystalline ferrites and PZT.  The following is added at the end of discussion part to address this comment.

The MEVC values, however, are smaller than the laminate counterparts made of polycrystalline ferrites and ferroelectrics. [25] In spite of the low MEVC values, the nanofibers are of importance for applications in miniature magnetic sensors for medical imaging, RF sensors, and for energy harvesting [39]. Efforts such as optimization of the ME response by enhancing the resistivity of the ferrite by annealing the fibers in oxygen rich atmosphere to eliminate the divalent Fe in order to decrease the leakage currents may be essential.

  1. Please check the minor issues in the manuscript, such as the subscript for Zn2Y on page 15, and providing high-quality images for Figures 4.

Response: We have corrected all the typos and provided a revised Figure 4.

Reviewer 2 Report

The authors prepared the nanofibers of Y- or W-type hexagonal ferrites, PZT or BTO, as well as the core-shell fibers of hexagonal ferrites/PZT (BTO) by elec-trospinning followed by sintering. The magnetization, polarization, and ME coupling are characterized. Most important is that a very large self-biased ME coupling was revealed. The work certainly contribute to the community interested in self-biased ME coupling of electrospun films, especially core-shell structure. However, I feel that the manuscript can be further improved by considering the following comment:

  1. The structure of the manuscript:

For the Section of Results and Discussions, the haxaferrite core- ferroelectric shell, or the ferroelectric core- haxaferrite shell need to be presented separately, and followed by the comparison of the two –structures and the underlying mechanism.

  1. Mechanisms for the self-biased ME coupling need to be discussed in more detail, as this is a very important experimental result.
  2. Some grammar mistakes need to be corrected, such as,

“Fibers of were free of impurities and characterization by magnetization, FMR, and magnetostriction …” (4th paragraph of page 2).

 “It was necessary use a heating and cooling rate of 0.5 ℃/min to avoid disintegration of the fibers.”

Only to name a few.

Author Response

Response to comments by Reviewer 2

We are grateful to the reviewer for the comments.  The manuscript has been revised by taking into consideration all of the comments. Our response to specific comments is given below.

“The authors prepared the nanofibers of Y- or W-type hexagonal ferrites, PZT or BTO, as well as the core-shell fibers of hexagonal ferrites/PZT (BTO) by electrospinning followed by sintering. The magnetization, polarization, and ME coupling are characterized. Most important is that a very large self-biased ME coupling was revealed. The work certainly contribute to the community interested in self-biased ME coupling of electrospun films, especially core-shell structure. However, I feel that the manuscript can be further improved by considering the following comment:”

  1. The structure of the manuscript:

For the Section of Results and Discussions, the haxaferrite core- ferroelectric shell, or the ferroelectric core- haxaferrite shell need to be presented separately, and followed by the comparison of the two –structures and the underlying mechanism.

Response:  We have separated the results on magneto-electric characterization into two parts as suggested.  Figures 11 and 12 are for ferrite core-ferroelectric shell and Figures 13 and 14 for ferroelectric cores and ferrite shells.  The ME response of these two different nano fibers is compared in the Discussion part.

  1. Mechanisms for the self-biased ME coupling need to be discussed in more detail, as this is a very important experimental result.

Response:  The following section is added to the Discussion part:

The most significant finding from the data in Figures 11 to 14 is the large MEVC at H=0 that could be attributed to the anisotropy field in the ferrite that acts as a built-in bias field. The low-frequency ME coefficient in the composite is directly proportional to the product of the piezomagnetic coefficient q = dl/dH (where l is the magnetostriction) and the piezoelectric coefficient d. In most of the ferromagnetic metals and spinel ferrites, l and q ≈ 0 at H = 0, resulting in MEVC = 0 at H=0.  ME response under zero external bias was reported in several systems, including layered composites with spinel ferrites and hexagonal ferrites [37,38]. In these composite systems, the remnant magnetization or the magnetocrystalline anisotropy acts as a bias field and, therefore, a nonzero q-value and MEVC.  In the present study, we estimated an anisotropy field Ha of 700 Oe in pure hexagonal ferrite fibers from FMR measurements.  In the core-shell fibers FMR data yielded a smaller Ha of 150-200 Oe that acts essentially as the bias field and resulting in a non-zero MEVC for H=0.

  1. Some grammar mistakes need to be corrected, such as,

“Fibers of were free of impurities and characterization by magnetization, FMR, and magnetostriction …” (4th paragraph of page 2).

 “It was necessary use a heating and cooling rate of 0.5 ℃/min to avoid disintegration of the fibers.”

Response:  We have corrected all the typos.

Reviewer 3 Report

This paper reports on the synthesis and the characterization ME properties in nanofibers of Y- and W-type hexagonal ferrites and composite core-shell fibers with ferroelectric lead zirconate titanate and barium titanate. The authors found taht magnetic and ferroelectric order parameters for the composite fibers were smaller than for pure ferrite and ferroelectric fibers. Furthermore, they achieved the highest ME coupling strengthen for naocomposite in this study. Detailed characterization and the achievement of the highest ME performance in the fibers deserve the publication in J. Compos. Sci. However, the following point must be addressed before accepting this paper. 

In Fig. 1, the authors show the expected XRD patterns of the respective ferrites. How did they obtain the patterns? Comparing the expected XRD patterns of Co2W and Zn2W, some differences are seen. For example, a peak around 23 deg. is only seen in the pattern of Co2W but not in Zn2W. Despite the same crystal structure, why the expected patterns are different? This must be clarified. 

Author Response

Response to comments by Reviewer 3

We thank the reviewer for the comments.  Our response to specific comments are given below.

“This paper reports on the synthesis and the characterization ME properties in nanofibers of Y- and W-type hexagonal ferrites and composite core-shell fibers with ferroelectric lead zirconate titanate and barium titanate. The authors found that magnetic and ferroelectric order parameters for the composite fibers were smaller than for pure ferrite and ferroelectric fibers. Furthermore, they achieved the highest ME coupling strengthen for naocomposite in this study. Detailed characterization and the achievement of the highest ME performance in the fibers deserve the publication in J. Compos. Sci. However, the following point must be addressed before accepting this paper. 

In Fig. 1, the authors show the expected XRD patterns of the respective ferrites. How did they obtain the patterns? Comparing the expected XRD patterns of Co2W and Zn2W, some differences are seen. For example, a peak around 23 deg. is only seen in the pattern of Co2W but not in Zn2W. Despite the same crystal structure, why the expected patterns are different? This must be clarified.”

Response: The comparison XRD patterns in Figure 1 are the XRD standard PDF card corresponding to the ceramic and are used as a reference for the patterns obtained for the fiber samples.

We added the following section to clarify the reviewer’s comment on XRD data for Co2W and Zn2W.

“Comparison of the XRD patterns for the fibers and the PDF standard in Figure 1 reveals that not all the peaks expected peaks are seen in the data for fibers, but some of the additional peaks are identified as impurity phases. In addition, factors such as stress on the nanofiber under test can lead to weakened peak intensities or deviations between the actual peak positions and the reference values. This is evident, for example, in the XRD data of Co2W and Zn2W nanofibers in which a peak around 23 deg is seen for Co2W, but not for Zn2W.”

Round 2

Reviewer 2 Report

The authors have revised their manuscript according to the reviewers' comments, and I suggest its acceptance in the present form.